# Mechanisms of Fetal Overgrowth in Gestational Diabetes: The Potential Role of SOCS2

**DOI:** 10.3390/nu17091519

**Published:** 2025-04-30

**Authors:** Luisa Hernández-Baraza, Yeray Brito-Casillas, Carmen Valverde-Tercedor, Carlota Recio, Leandro Fernández-Pérez, Borja Guerra, Ana M. Wägner

**Affiliations:** 1Instituto Universitario de Investigaciones Biomédicas y Sanitarias, Universidad de Las Palmas de Gran Canaria, 35016 Las Palmas de Gran Canaria, Spain; luisa.hernandez@ulpgc.es (L.H.-B.); carmen.valverde@ulpgc.es (C.V.-T.); carlota.recio@ulpgc.es (C.R.); leandrofco.fernandez@ulpgc.es (L.F.-P.); ana.wagner@ulpgc.es (A.M.W.); 2Department of Endocrinology and Nutrition, Complejo Hospitalario Universitario Insular Materno-Infantil, 35016 Las Palmas de Gran Canaria, Spain

**Keywords:** SOCS2, gestational diabetes mellitus, macrosomia, fetal growth

## Abstract

During pregnancy, the maternal body adapts in several ways to create an optimal environment for embryonic growth. These changes include endocrine and metabolic shifts that can lead to insulin resistance and gestational diabetes mellitus (GDM), impacting both the mother and fetus in the short and long term. Fetal macrosomia, a condition where the fetus is significantly larger than average, is a primary concern associated with GDM. Although the underlying mechanism remains unclear, a pregnancy-induced proinflammatory state, combined with altered glucose homeostasis, plays a critical role. Several cytokines and hormones, such as interleukin 6 (IL-6), insulin growth factor 1 (IGF-1), prolactin (PRL), or progesterone, are essential for fetal growth, the control of the inflammatory response, and the regulation of lipid and carbohydrate metabolism to meet energy demands during pregnancy. However, although the role of these cytokines in metabolism and body growth during adulthood has been extensively studied, their implication in the pathophysiology of GDM and macrosomia is not well understood. Here, we review this pathophysiology and pose the hypothesis that an aberrant response to cytokine receptor activation, particularly involving the suppressor of cytokine signaling 2 (SOCS2), contributes to GDM and fetal macrosomia. This novel perspective suggests an unexplored mechanism by which SOCS2 dysregulation could impact pregnancy outcomes.

## 1. Introduction

Suppressors of cytokine signaling (SOCS) proteins act as key regulators of cytokine and growth factor signaling through the Janus kinase (JAK)/signal transducers and activators of transcription (STAT) pathway. The SOCS family comprises eight proteins: SOCS1-7 plus the cytokine-inducible SH2-containing protein (CISH) [1,2,3]. Specifically, SOCS2 is prominently expressed in the placenta [4], particularly in endothelial cells and the placental vasculature. SOCS2 modulates the interaction of placental villi with adjacent cells and acts as a negative regulator of proinflammatory cytokine responses [5].

Given the multitude of cytokines and growth factors that signal through the JAK/STAT pathway, SOCS2 emerges as a potentially significant yet underexplored protein involved in maintaining pregnancy homeostasis. The primary functions of SOCS2 are associated with body growth, the inflammatory process, and cytokine-mediated lipid and carbohydrate metabolism (Figure 1) [1,2,3]. Despite its importance, the specific role of SOCS2 in the physiology of normal pregnancy remains inadequately understood and further investigation into its interactions with cytokines, growth factors, and hormones during pregnancy is needed [6,7,8].

Understanding pregnancy and its complications is crucial in clinical practice. However, in-depth studies are often hindered by the complexities of replicating embryonic development and acquiring placental and embryonic tissues. The mechanisms behind complications such as gestational diabetes mellitus (GDM) remain poorly understood, with controversial hypotheses surrounding its etiology and prognosis [9,10,11]. One important effect associated with GDM is fetal macrosomia, which poses perinatal risks to both maternal and fetal health, with long-term consequences on infant metabolism. Although there are numerous meta-analyses of macrosomic newborns, studies on the biomolecular mechanisms involved are sparse [12].

The mechanisms of the maternal–fetal interaction through the placenta, as well as complications arising during pregnancy, such as metabolic disorders during or before gestation and their potential influence on growth disorders of the offspring, are still not fully understood. Here, we review the mechanisms involved in pregnancy and their association with complications, specifically GDM and related macrosomia. We also propose the SOCS2 protein as a focus for research into the physiology and physiopathology of pregnancy, as dysregulation in its signaling pathway may lead to clinical consequences, such as insulin resistance and fetal overgrowth.

## 2. Endocrine–Metabolic Sketch of Pregnancy

Pregnancy is an energetically demanding process that relies on complex endocrine–metabolic adaptations, requiring synchronized interactions between the mother and the fetus. Throughout pregnancy, an elevated maternal metabolic rate supports fetal growth and development and increased energy requirements [13,14]. In early and mid-pregnancy, placental hormones and growth factors promote fat storage, which serves as a necessary energy source for the fetus later in pregnancy. During the third trimester, lipolysis of expanded fat stores results in increased levels of plasmatic fatty acids and glucose, with lipids becoming the primary energy source for maternal needs, while glucose is preserved for the developing fetus [14].

Glucose homeostasis during pregnancy is crucial because the fetus relies entirely on maternal blood and the placenta for its glucose supply [14]. In the first trimester, fetal glucose demand is primarily met through the regulation of maternal hormones, including estrogens, insulin, and cortisol, which decrease maternal energy expenditure, delay blood glucose clearance, and promote fat storage [13,15]. As pregnancy progresses, the fetal glucose demand increases, necessitating enhanced maternal hepatic gluconeogenesis and reduced insulin sensitivity [13,16]. Maternal endocrine and metabolic adaptations include insulin resistance, increased β-cell expansion and response, and a moderate rise in blood glucose levels after food intake. Changes in circulating free fatty acids (FFAs), triglycerides (TGs), cholesterol, and phospholipids ensure adequate nutrient supply for the developing fetus [14,17].

Endocrine and metabolic changes during pregnancy primarily result from hormonal signaling within the feto-placental unit (FPU). Consequently, maternal adaptations to these pregnancy-related changes directly impact fetal and placental development [18]. However, in GDM pregnancies, these adaptations may be disrupted, leading to adverse pregnancy outcomes.

The primary physiological changes in pregnancy stem from increased levels of maternal hormones (Figure 2), particularly progesterone and PRL, which are essential for meeting the energy requirements for embryo implantation. PRL provides luteotropic support to the corpus luteum and is later replaced by placental lactogen (PL), which elevates maternal glucose, encourages lipolysis, and induces insulin [19]. Additionally, maternal β-cell proliferation is regulated by these hormones and insulin secretion is enhanced to accommodate increased demands and resistance, notably in the third trimester [16,20].

Progesterone and estrogens orchestrate the female reproductive cycle and prepare the endometrium for embryo implantation. Progesterone is crucial for oocyte maturation, pregnancy progression, and labor, while estrogens are moderated early in pregnancy to prevent uterine growth [17]. Both hormones influence dietary intake and glucose balance during pregnancy. Estrogens promote insulin sensitivity, increasing insulin levels and activity, while progesterone reduces insulin efficacy and glucose transport, potentially leading to glucose intolerance [14,21].

Ghrelin and leptin, hormones secreted by the placenta, meet the nutritional needs of the developing fetus. Ghrelin stimulates appetite, promoting increased food intake, while leptin reduces feeding behavior and enhances energy expenditure [13,22]. In this way, a state of hyperleptinemia is followed by leptin resistance, which helps to maintain increased energy intake, adipose storage, and lipid mobilization [13,14,23].

The adipokine adiponectin and the myokine irisin play roles in glucose metabolism and insulin resistance, with adiponectin stimulating fatty acid oxidation (FAO) and regulating glucose metabolism, and irisin improving glucose homeostasis by reducing insulin resistance and enhancing lipid profiles [24].

Other growth hormones also appear during pregnancy. The maternal pituitary growth hormone (GH) is progressively supplanted by the placental growth hormone (pGH), which completely takes over by the second trimester. pGH adjusts to maternal circulating glucose and insulin levels to ensure efficient glucose transfer to the fetus, enhancing nutrient availability. This regulation of pGH levels occurs through direct interaction with the GH receptor (GHR) or indirectly via IGF-1. pGH plays significant roles in placental development, maternal adaptation, and fetal growth by modulating IGF-1 levels, which increase proportionally throughout pregnancy and decrease post-delivery, when maternal GH re-emerges [24,25,26,27,28].

The placenta begins to form after conception and develops over the first three months, growing with the uterus from the fourth month onward [29]. Initially, it produces human chorionic gonadotropin (hCG) to stimulate early embryonic development, peaking around week 10 before decreasing. It binds to the thyroid-stimulating hormone (TSH) receptor, thereby stimulating the thyroid gland, and reducing TSH secretion, which may correspond to the gradual decline in free thyroxine (T4) and triiodothyronine (T3) hormones starting from the second trimester [30].

The maternal hypothalamus–pituitary–adrenal (HPA) axis changes during pregnancy. The placenta synthesizes the corticotropin-releasing hormone (CRH), which is involved in the timing of birth, facilitates glucose transport to placental cells and the fetus, and affects the mother’s physiological state. Despite increased placental CRH levels, adrenocorticotropic hormone (ACTH) and cortisol levels rise moderately because the maternal anterior pituitary becomes less responsive to CRH stimulation, preventing excessive increases [31]. Excessive physiological hypercortisolism may lead to insulin resistance by reducing the insulin receptor substrate 1 (IRS-1) content [14,32].

Finally, oxytocin accumulates in the posterior pituitary gland, and its release inhibited to prevent premature labor. At full term, oxytocin is fully released, preparing the mother for childbirth and lactation [17].

Several of the main hormones involved in pregnancy, such as PRL, PL, progesterone, estrogens, GH, pGH, IGF-1, insulin, and leptin, are transduced by the JAK/STAT pathway. Therefore, SOCS2, as a key regulator of this pathway, may be involved in the endocrine and metabolic adaptations during pregnancy [1,3,33].

## 3. Embryonic and Fetal Growth

Embryonic and fetal growth involve a complex interplay among the mother, fetus, and placenta to meet nutritional demands.

The placenta, a fetal organ with both endocrine and immune functions, facilitates the transport of nutrients from maternal blood and produces numerous mediators involved in fetal growth. Receptors on the placental membrane enable direct interaction with growth and metabolic pathways [16,34,35,36,37], and disruptions in these receptors or hormone levels can lead to growth disorders [12,15,38].

Fetal growth strategies differ between sexes, with females being more sensitive to the maternal uterine environment, while males prioritize growth, leading to higher risks of gestational complications. An X-linked genetic component determines sexual dimorphism in the placenta and early fetal development [39,40]. Regulatory factors and fetal hormone secretion activate sexually dimorphic genes, contributing to variations in fetal growth. Sex-specific epigenetic mechanisms impact the placenta, with differences in placental glucose, amino acid transporters, and fatty acid composition between sexes [40,41,42]. Moreover, in cases of maternal GDM and obesity, sex-specific differences are observed in inflammatory cytokines, amino acid and lipid transport, as well as microRNA (miRNA) expression, which can potentially explain variations in fetal growth and outcomes [40,42].

The IGF-1 pathway is crucial for fetal growth, enhancing protein and glucose synthesis, and promoting maturation [12,15,43,44] (Figure 3). IGF-1, along with adiponectin, orchestrates the downstream mammalian target of the rapamycin (mTOR) pathway, which is essential for regulating lipids, amino acids, folate, and glucose through the placenta [12,44].

Disruption in the IGF-1 signaling axis can lead to growth disorders [15,37,45] (Figure 3). The IGF system is the most critical endocrine and paracrine pathway for both fetal and placental growth [15,34,44], with IGF-1 and IGF-2 ligands expressed in fetal tissues [34,44]. Maternal IGF-1 cannot cross the placental barrier [46], prompting some authors to question its primary impact on fetal growth and consider GH and its receptor GHR instead [47,48,49]. The interplay between fetal and maternal IGF-1 is indeed crucial for fetal development. Maternal IGF-1 facilitates nutrient transfer to the fetus, supporting its growth and development. Conversely, fetal IGF-1 drives growth by promoting anabolic processes and DNA synthesis [50,51]. The placental barrier plays a significant role in regulating this process. By expressing IGF-1 receptors and IGF-binding proteins (IGFBPs), the placenta modulates the availability and function of IGF-1 between the mother and fetus [50,52]. This regulation is essential to ensure proper nutrient delivery and growth signals, although it can also pose limitations when placental function is compromised.

In vivo studies in rats show increased IGF-1 mRNA alongside detectable GHR mRNA and immunoreactivity in embryos. Furthermore, in vitro experiments confirmed GHR expression, during organogenesis [47,53]. Disruptions in the GHR-JAK2-STAT5-SOCS2 signaling pathway are associated with somatic growth disorders, gender dimorphism, liver disease, and insulin resistance [15,33]. Nevertheless, key studies refute the need for GH for prenatal growth, as in vivo models lacking GHR show no growth disorders at birth [54,55].

## 4. Gestational Diabetes

The American Diabetes Association (ADA) defines GDM as any degree of glucose intolerance with onset or first recognition during pregnancy [56]. This condition commonly emerges in the second or third trimester and affects up to 16% of pregnancies. The prevalence of GDM is rising, paralleling increases in maternal obesity, overweight, and age. GDM is linked to immediate and long-term adverse outcomes for the mother and child. These include an increased likelihood of birth trauma and shoulder dystocia due to neonatal macrosomia, as well as respiratory distress syndrome, neonatal hypoglycemia, hyperbilirubinemia, elevated fat mass, cardiometabolic disease, and a higher incidence of obesity extending into adolescence.

Women with GDM face a greater risk of preeclampsia, labor induction, preterm birth, cesarean delivery, metabolic syndrome, type 2 diabetes (T2D), and renal and cardiovascular diseases [21,53,57,58,59,60] (Figure 4). Therefore, improving our understanding of GDM prevention, early detection, and treatment is of paramount importance.

Normal pregnancies are marked by an increase in glucose-stimulated insulin secretion and maternal pancreatic β-cell proliferation. The expansion of β-cell mass and its enhanced insulin secretion capacity are induced by both maternal hormones, such as PRL, and placental hormones, such as PL. This increase in insulin production protects against the glucose intolerance encountered during pregnancy. In pancreatic islets, PRL/PL signaling triggers the phosphorylation and nuclear translocation of STAT5, which in turn boosts β-cell proliferation. GDM arises when insulin resistance cannot be compensated by maternal insulin secretion [53,61]. Interestingly, in rodents, there is a surge in β-cell proliferation during the second third of gestation to meet the increased insulin demand [16,61]. Although the exact molecular mechanisms of GDM remain unclear, it is hypothesized that a maladaptation of β-cells to pregnancy-induced glucose intolerance may underlie the condition [16,61,62]. However, this process is not straightforward, as numerous other factors also play a role and can alter placental function in cases of hyperglycemia [14,38].

Animal model studies are essential for understanding the mechanisms of fetal development and gestation, but they have significant limitations. The placental interface differs between species; in humans, maternal blood directly contacts a single layer of trophoblasts in the chorion plate, whereas in rodents, the placental barrier is hemotrichorial, consisting of three layers of trophoblasts. Additionally, rodent pregnancies are typically multiparous, and their gestation period is much shorter than that of humans. Therefore, rodent models are valuable for studying basic biological processes, but their translatability to human pregnancies should be carefully considered [63].

The placenta has a homeostatic capacity, responding to maternal metabolic perturbations to stabilize the fetal environment. However, in cases of obesity and GDM, the protective effects of the placenta may be overwhelmed by an excessive metabolic burden from maternal overfeeding [64]. Maternal obesity and GDM are often linked to a chronic, low-grade inflammatory state known as ‘metainflammation’, which is distinct from acute inflammation. This inflammatory environment may be one of the mechanisms that predispose offspring to metabolic disorders later in life [65,66,67].

The limited capacity of the placenta to tolerate and respond to environmental changes and stresses, particularly in cases of maternal obesity and GDM, involves several key factors. Proinflammatory cytokines have been identified as crucial signals that induce the up-regulation of endothelial lipase (EL). Additionally, the placenta exhibits a limited ability to store triglycerides and an overall reduced capacity to maintain homeostasis and regulate substrate supply to the fetus under increased metabolic load [64,68].

Normal pregnancy involves an altered inflammatory profile compared to the non-pregnant state, with a tightly regulated balance between pro- and anti-inflammatory cytokines likely necessary for normal implantation, trophoblast invasion, and placentation [65]. However, increased adipose tissue in obese women leads to heightened inflammation in both placenta and maternal adipose tissue. The latter, functioning as an endocrine organ, secretes adipokines and cytokines, and except for adiponectin, the placenta secretes a similar profile of cytokines, leading to metainflammation in obese and GDM mothers [67].

In the placentas of obese women, the levels of CD14 and CD58 macrophages are significantly higher, with increased levels of pro-inflammatory cytokines such as IL-1, IL-6, and tumor necrosis factor (TNF-α). This inflammation may be the primary mechanism underlying the increased insulin resistance observed in connection with obesity and GDM [67]. Additionally, in pregnancies of obese mothers, changes in placental metabolism are evident in the metabolites of key signaling pathways, such as mTOR, IGF-1, adipokine signaling, and nutrient transport [12]. The transplacental flow of nutrients, including free fatty acids and amino acids, is altered and has been linked to both GDM and fetal growth. These changes are positively correlated with maternal body weight and body mass index (BMI), and obese or overweight mothers are more likely to develop GDM [69]. In contrast, E2 plays a protective role in pregnancy by modulating the cytokine profile. It inhibits Th1 proinflammatory cytokines, such as IL-12, TNF-α, and interferon (IFN-γ), while stimulating Th2 anti-inflammatory cytokines, including IL-10, IL-4, and TGF-β [70]. However, in obese pregnancies, the protective effects of E2 are reduced, leading to an imbalance in immune regulation and increased inflammation. This reduction in the modulatory effects induced by E2 may contribute to the heightened risk of complications associated with maternal obesity and GDM [71].

## 5. Macrosomia

The terms ‘large for gestational age’ (LGA) and ‘macrosomia’ describe newborns who are significantly larger than average at birth. However, LGA refers to infants whose birth weight is above the 90th percentile for their gestational age and sex. The ADA defines macrosomia as having ‘abnormally large babies, often born to women with diabetes’ [10]. Macrosomia is typically defined by a birth weight exceeding 4000 g and is associated with various short- and long-term health risks for both the mother and child. Its incidence is rising globally [11,60]. Maternal complications include cesarean delivery, extended labor, postpartum hemorrhage, soft tissue injuries, and pre-eclampsia. Newborns may face immediate risk of shoulder dystocia, respiratory distress, neonatal hypoglycemia, and birth injuries, and long-term risk of obesity, diabetes [11,60], psychiatric disorders [72], dental anomalies (especially in the maxillofacial area) [73], and cardiovascular disease [21]. Research indicates that macrosomic infants born to GDM mothers are at a higher risk of adverse perinatal and postnatal outcomes compared to those born to non-GDM mothers [74].

Efforts to prevent fetal overgrowth have shown limited success. While enhanced glycemic control has been effective in reducing macrosomia rates, the detailed molecular and cellular mechanisms behind fetal macrosomia remain elusive. Overnutrition through the placenta is linked to macrosomia; with diabetes, increased maternal nutrient availability leads to a rise in leptin receptor expression in the fetus [75]. Moreover, excessive glucose transfer across the placenta in cases of GDM triggers fetal insulin secretion, which causes hypertrophy in insulin-sensitive tissues such as adipose and hepatic tissues, skeletal and myocardial muscle, and the islets of Langerhans, thus promoting rapid growth [14,76].

Fetal growth is regulated by the hormonal systems of the fetus, mother, and placenta [15,47]. A hyperglycemic maternal environment can modify hormone levels, alter the fetal epigenome, and increase fetal fat mass, raising the risk of childbirth complications and future metabolic syndromes [14,37,45]. FFA and amino acids, part of the transplacental nutrient flow, correlate with maternal body weight and BMI and are associated with GDM [57,69]. Elevated levels of branched-chain amino acids (BCAAs) like leucine, valine, and isoleucine, as well as aromatic amino acids (AAAs) like tyrosine and phenylalanine, have previously been associated with T2D due to their role in insulin resistance. The accumulation of amino acids in plasma indicates the dominance of amino acid production (from diet and proteolysis) over clearance. Since insulin promotes protein synthesis and inhibits proteolysis, we can predict that states of insulin deficiency or insulin resistance would be associated with hyperaminoacidemia [77]. This accumulation of BCAAs and AAAs in plasma further exacerbates insulin resistance by interfering with insulin signaling pathways. These amino acids have been shown to have a similar effect during pregnancy, where their increase in maternal blood promotes β-cell replication and leads to excessive insulin secretion, thereby impairing amino acid and energy metabolism [78,79]. Additionally, circulating FFAs in maternal blood, which pass through the placenta, contribute to the development of insulin resistance and fetal macrosomia [14,80].

The mTOR pathway is modulated by adiponectin and IGF-1 during non-pathological pregnancies. However, in GDM and obese pregnancies, lower circulating levels of adiponectin are observed, which correlate positively with fetal overgrowth [12,81,82], suggesting another disrupted metabolic pathway contributing to fetal macrosomia. The mTOR signaling pathway is known for its role in obesity, a risk factor for diabetes, which also influences embryonic growth [83,84]. Rodent models of GDM and obesity during pregnancy have shown an activation of placental mTOR complex1 (mTORC1), which acts as a positive regulator of placental amino acid transporters, thereby increasing fetal growth [44,83]. In vitro studies on placental explants have indicated that high glucose levels decrease mitochondrial fatty acid oxidation and increase triglyceride accumulation, which is accompanied by overexpressed lipases and lipid transfer proteins, which induce the placental IGF-1 signaling pathway [38,85,86]. Elevated leptin levels have been observed in pregnant women with GDM [87], but insulin treatment has been reported to normalize these values in the placenta. Furthermore, estrogen is known to improve placental insulin sensitivity through its interaction with the ER, and it has been suggested that ER-mediated mechanisms are altered in GDM [75,88].

Studies comparing the placentas of women with and without GDM highlight the role of IGF-1 signaling in macrosomia [15,34,35,43]. The IGF-1 signaling pathway is amplified in women with obesity and GDM and has been positively associated with macrosomia [12,15,43,44]. The insulin receptor (IR) protein was reported to be increased in the placentas of women with GDM [35,75,89]. The activated IR phosphorylates the IRS, which acts as a docking protein for other signaling proteins, thereby activating various metabolic and cellular pathways. Indeed, the stimulation of phosphorylated IRS has also been reported in the placentas of women with GDM [75,90].

Overweight and obesity are linked to fetal macrosomia due to an environment rich in proinflammatory cytokines, growth factors, and hormones that influence growth [69,91]. An inflammatory profile enhances glucose absorption by the placenta [91], which can lead to macrosomia. When GDM or maternal obesity is present, there is an increase in placental proinflammatory cytokine levels; this exposes the fetus to TNF-α, monocyte chemotactic protein-1 (MCP-1), IL-1, and IL-6, among others. These factors have been associated with delivering larger infants [83].

Fetal metabolic programming caused by an adverse intrauterine environment can lead to metabolic syndrome in adult offspring. Although the precise mechanisms remain incompletely understood, epigenetic programming is proposed as a critical underlying factor. This process influences gene expression and cellular function through stable epigenetic modifications that do not alter the DNA sequence. These modifications can be maintained through mitosis and transmitted across generations, thereby contributing to metabolic syndrome in adults [92]. Genes associated with adipose tissue development, hyperphagia, energy balance regulation, lipid metabolism, insulin signaling, adipocytokines, proinflammatory factors, pancreatic islet beta cell development, and DNA or histone-modifying enzymes can undergo changes due to fetal epigenetic modifications. This contributes to metabolic syndrome in adults. Notable genes affected include IGF1R, IGF2, FGF21, ER, Glut4, IGFBP1, leptin, and adiponectin [51,92,93].

Thus, understanding the mechanisms and factors that regulate fetal growth is crucial for preventing macrosomia and its related complications.

## 6. Biomarkers of Gestational Diabetes and Macrosomia

The United States Food and Drug Administration (FDA) defines a biomarker as a “defined characteristic that is measured as an indicator of normal biological processes, pathogenic processes, or responses to an exposure or intervention” [94]. Biomarkers are objective and measurable signs of biological activity and are crucial in clinical practice. They offer insights into morbidity and mortality and aid in disease prevention, treatment personalization, monitoring of disease progression, and evaluation of treatment effectiveness. Importantly, they help us understand the mechanisms behind certain disorders [94].

Common clinical biomarkers for predicting an increased risk of GDM include maternal age, family history of T2D, and pregestational overweight or obesity [11,67]. GDM is linked to a higher risk of fetal macrosomia and neonatal hypoglycemia, which occurs more frequently among infants born to GDM mothers [11,67]. Extensive research has been conducted on the cellular and metabolic pathways involved in pregnancy and GDM to identify key metabolites associated with the pathogenesis of GDM and macrosomia, which may be useful diagnostic and preventive biomarkers. The biomarkers identified in the literature related to GDM and macrosomia, found in both placental tissue and maternal blood, are discussed here and detailed in Table 1.

### 6.1. Gestational Diabetes Mellitus Biomarkers

a.Maternal serum biomarkers

The maternal lipid profile has been extensively studied, yet much remains to be explored regarding its influence on fetal growth [14,95,96]. Serum TG, FFA, and cholesterol concentrations are comparable in pregnancies of GDM and normoglycemic mothers. However, in cord blood, TG and FFA levels are higher in GDM and correlate with newborn weight and fat mass [97]. FAO is reduced in placentas and associated with increased levels of non-esterified fatty acids (NEFAs) in the serum of GDM mothers. Interestingly, FAO inhibition is believed to be compensated by the esterification pathway [95].

Recently, Mingjuan Luo et al. [98] identified TG and cholesterol as independent risk factors for pregnancy outcomes in GDM mothers. Furthermore, they propose other key lipid metabolites—such as γ-linolenic acid, heptadecanoic acid, oleic acid, palmitic acid, and palmitoleic acid—as potential biomarkers of pregnancy complications. These metabolites primarily participate in fatty acid biosynthesis [98]. Although diabetes during pregnancy does not significantly alter the mother’s lipid profile, it does impact that of the fetus. Enhanced expression of lipid pathway genes and lipogenesis proteins has been identified in placentas from GDM cases, which may explain variations in fetal fat mass [99].

Many biomarkers are related to the inflammatory response observed in obesity [12,14,58,100]. Leptin levels interfere with maternal glucose metabolism and are correlated with glycated hemoglobin (HbA1c) and newborn weight. Other studies link inflammatory agents, adipokines, and cell adhesion molecules—such as intercellular adhesion molecule 1 (ICAM-1) or vascular cell adhesion molecule 1 (VCAM-1)—with insulin resistance and hyperglycemia during pregnancy [58]. Additionally, elevated levels of the proinflammatory cytokine TNF-α [58] and fibroblast growth factor 21 (FGF21) have been detected in GDM pregnancies. FGF21 plays a significant role in glucose and lipid homeostasis; itincreases in obesity and T2D [101,102].

Amino acid metabolism is also disrupted in GDM mothers. Elevated levels of BCAA, such as leucine, isoleucine, and valine, have been linked to insulin resistance in GDM [57]. Due to the inhibition of gluconeogenesis in GDM, concentrations of certain gluconeogenic amino acids like glutamine and glutamate are increased during GDM pregnancies [57,103]. The role of amino acids in GDM has been further explored through studies of maternal urine, which identified additional metabolites associated with an increased risk of GDM, including ethanolamine and 1,3-diphosphoglycerate. Moreover, higher plasma concentrations of 2-hydroxybutyrate, succinic acid, and 3-hydroxybutyric acid have been observed in GDM mothers. These findings suggest a disturbance in the Krebs cycle that contributes to impaired glucose homeostasis and insulin resistance [11,57,60,104].

Circulating levels of pregnancy hormones have also been investigated as potential markers of GDM. Mutations in PRL-GH genes and their receptors—such as reduced PRLR signaling in beta-cells—have been identified in GDM mothers [105,106]. During normal pregnancies, steroid hormones are responsible for regulating glucose homeostasis, insulin secretion, appetite control, and lipid metabolism to support the necessary metabolic changes [107]. However, GDM pregnancies are characterized by a reduction in plasma E2 levels [71,108]. Additionally, the placenta secretes neuroactive hormones [109] like kisspeptin, a decrease of which during pregnancy accommodates maternal physiological adaptations [107,110].

b.Placental and cord blood biomarkers

The placenta functions as both a connector and a barrier between the maternal and fetal environments, catering to the embryo’s requirements for nutrient transport, immune response, metabolism, and growth [91,95,111]. Placentas tend to be larger and structurally distinct in GDM than in normoglycemic mothers [75,91]. These differences could serve as potential targets for the prevention or management of GDM and macrosomia.

Increased glucose uptake in GDM placentas reduces mitochondrial FAO capacity and enhances TG accumulation [85,95,112]. Changes in placental metabolism due to GDM are associated with alterations in the metabolites of key placental signaling pathways [12].

Maternal pituitary GH is suppressed due to its replacement by the pGH secreted into the circulation, which in turn regulates maternal serum IGF-1 [113,114,115]. This indirect effect impacts fetal growth. Recently, the IGF-binding protein 1 (IGFBP-1) gene has been found to be deficiently expressed in the placenta in GDM pregnancies, while it is up-regulated in normal pregnancies [116]. Furthermore, the overactivation of IGF-1 is associated with an increase in other metabolites, such as IRS-1 and the phosphatidylinositol-3-kinase subunit (p58α), which are found at high levels in the placenta of GDM pregnancies—even in GDM mothers with well-controlled glucose [44,75]. The increase in maternal adipocytes stimulates the placenta to release proteins involved in insulin resistance and impaired glucose tolerance, which has been associated with GDM [22].

Transplacental nutrition may also be affected by the increased circulation of amino acids in GDM, which also stimulates the nutrient-IGF1 axis and transporters. One such transporter is the sodium-coupled neutral amino acid transporter 1 (SNAT1), which has been detected in placentas from GDM mothers with macrosomic offspring [15,44,75].

The levels of ERα were also found to be higher in placentas from GDM than normoglycemic mothers, although no information was provided on newborn weight [75]. Conversely, a decrease in placental AMP-activated protein kinase (AMPK) activity was observed in GDM mothers. This decrease was associated with increased maternal serum and placental IGF-1 levels and correlated with newborn weight [44].

Alonso et al., 2006 [75], proposed that the primary function of insulin in the placenta was to stimulate mitogenic activity to support the metabolic demands of a ‘diabetic’ placenta. Additionally, insulin enhances placental signaling to shield the fetus from hyperglycemic spikes that may be associated with GDM [75,117]. Additionally, increased downstream phosphorylation of phosphatidylinositol-3 kinase (PI3K), AKT, and glucose transporter-4 (GLUT4) during GDM pregnancies may intensify insulin resistance due to impaired insulin signaling [38,111].

Recent studies have discovered gender differences in placental biomarkers related to GDM and long-term health consequences for offspring [118,119,120]. Male fetuses may be more susceptible to maternal hyperglycemia than female fetuses, which can be attributed to their quicker growth rates, higher demand for nutrients (particularly glucose), and greater insulin sensitivity. This gender-specific link suggests that GDM is a risk factor for obesity in male offspring but not in females. Conversely, maternal BMI is an indicator of adiposity in female newborns but not males [119]. Furthermore, biomarkers that regulate appetite and energy metabolism during GDM pregnancies also exhibit gender disparities. For instance, insulin, C-peptide, leptin, and various growth factors are significantly higher in the cord blood of boys from mothers with GDM; however, only IGF-1 was detected in girls [118,120]. These findings suggest that male and female fetuses experience different intrauterine environmental programming, which may lead to lifelong epigenetic changes.

### 6.2. Macrosomia Biomarkers

a.Maternal serum biomarkers

The identification of targets and biomarkers for macrosomia is closely linked to maternal factors. Key clinical characteristics associated with prenatal macrosomia diagnosis include pregestational obesity, GDM, weight gain during pregnancy, glucose-lowering treatment for pre-existing diabetes, multiparity, and older age.

Altered metabolism and hormone levels in GDM mothers contribute to excessive fetal growth by enhancing transplacental nutrient transfer through IGF-1 mechanisms [44,100]. Poor glycemic control during pregnancy leads to adipose tissue accumulation in newborns from GDM mothers [14,121]. C-peptide is a marker studied for macrosomia, since it has been previously reported to be associated with LGA newborns [86]. Conversely, low circulating adiponectin in maternal serum, a feature of obesity and GDM, stimulates fetal growth by activating the placental insulin/IGF-1/mTOR signaling pathway.

Placental GH regulates maternal IGF-1 levels [54,107,115], which decreases IR expression and signaling [89]. Since maternal IGF-1 signaling promotes fetal growth through nutrient transport mechanisms [46], an imbalance in placental GH may be implicated in conditions of fetal overgrowth.

b.Placental and cord blood biomarkers

The placenta is a critical tissue for understanding fetal growth mechanisms and the impact of a hyperglycemic environment. Since macrosomia results in an augmented fetal fat mass, the transfer of lipids through the placenta has also been investigated. The placental angiopoietin-like proteins, ANGPTL3, ANGPTL4, and ANGPTL8, regulate transplacental nutrition and positively correlate with birth weight in GDM pregnancies [122].

Maternal and fetal IGF-1 are regulated by a positive feedback loop. Maternal IGF-1 signaling stimulates fetal growth through increased amino acid transport, inducing overnutrition and activating the fetal IGF-1 axis [44,46], whose components have been correlated with newborn birth weight [44,123]. 

In contrast, in placentas from pregnancies affected by intrauterine growth restriction (IUGR), an increase in activated IR and total non-activated IRS-1 has been reported, as well as higher levels of the pro-inflammatory cytokine IL-6, which correlates with decreased levels of protein kinase B (AKT) and adiponectin. An upsurge in the SOCS2 protein has also been observed, which inversely correlates with all measured growth parameters. This could be attributed to the role of SOCS2 as both a direct and indirect negative regulator of the IR, IRS-1, and GHR signaling pathways. This indicates an interaction between SOCS2 activity, insulin action, and growth [35].

Although there is evidence of biomarkers associated with the insulin pathway, some researchers challenge the view that the insulin signaling pathway is the primary factor in macrosomia development and instead propose alternative mechanisms for fetal growth in GDM and normoglycemic mothers [38,111]. For instance, a 2018 study by W. Zhang et al. [38] reported elevated levels of Rho guanine nucleotide exchange factor 11 (ARHGEF11), an inhibitor of the IGF-1 pathway, which could contribute to macrosomia in normoglycemic pregnancies [38].

Epigenetic biomarkers of placental growth have been identified. For instance, miR-21, a miRNA targeting the JAK-STAT signaling pathway, is overexpressed in the placentas of newborns with macrosomia [53,124,125]. Additionally, other miRNAs, such as miR-675, have been implicated in embryonic growth [53,126]. These findings suggest that determining and comparing the expression levels of these crucial epigenetic biomarkers in GDM vs. normoglycemic could be clinically relevant [53].

The clinical applicability of biomarkers for gestational diabetes mellitus (GDM) and macrosomia holds promise for enhancing prenatal care and improving pregnancy outcomes. Given the challenges in accessing the placenta for diagnosing GDM or macrosomia during pregnancy, maternal serum biomarkers emerge as potential diagnostic candidates for these conditions. Biomarkers such as C-peptide, adiponectin, and insulin-like growth factor 1 (IGF-1) provide valuable insights into fetal growth patterns associated with GDM.

A better understanding of the causes of GDM, macrosomia, and maternal–fetal interactions via the placenta will identify new markers to elucidate these mechanisms. This knowledge can be leveraged to prevent disorders associated with development and pregnancy.

**Table 1 nutrients-17-01519-t001:** Biomarkers of gestational diabetes mellitus (GDM) and macrosomia as reported in the literature.

Biomarker	Origin	Up (↑)/Down (↓) Regulation	Role/Function	Type of Study	Evidence
2-Hydroxybutyric ^##^ 3-Hydroxybutyric ^##^	Maternal	↑	Ketone body derived from fatty acid oxidation	Metabolic plasma analysis (maternal blood samples)	[57,104]
Adiponectin ^#&^	Maternal	↓	Stimulation of FAO, enhancement glucose metabolism and reduction in insulin resistance throughout	Primary cell culture of human placentas/Human placenta and maternal blood samples	[22,82]
AMPK ^&&^	Placental	↓	Activation of cellular metabolism	Human placenta and maternal blood samples	[44]
ANGPL-3-4-8 ^#&^	Placental	↑	Increased transplacental nutrition and lipid and glucose metabolism	Maternal and cord blood and human placenta samples	[122]
ARHGEF11 ^##^	Placental	↑	Activation of G protein signaling and cellular processes such as insulin secretion, insulin signaling, and lipid metabolism	Human placenta samples	[38]
BCAA ^##^ Valine, Isoleucine, Leucine	Maternal	↑	Essential amino acids that cannot be produced by the body and must be primarily obtained from the diet. Related with GDM.	Metabolic plasma analysis (maternal blood samples)	[57]
C-peptide ^#&^	Maternal	↑	Marker of endogenous insulin secretion. Derived from the cleavage of proinsulin in insulin.	Maternal and cord blood samples	[86]
E2 ^##^	Maternal/Cord blood	↓	Sex female steroid that maintains β-cell survival and glucose homeostasis	Clinical, anthropometrical data and maternal blood samples/Cord blood samples	[71,108]
EL	Placental	↑	Mediation of the uptake of fatty acids from high-density lipoproteins	Human placenta samples	[68]
ER-α ^$$^	Placental	↑	Enhancement glucose uptake capacity	Human placenta samples	[75]
FAO ^#&^	Placental	↓	Breakdown of fatty acids into CoA for use as an energy resource	Human placenta and maternal blood samples	[85,95]
FFA ^#&^	Cord Blood/Maternal	↑/=	Source of energy for the body	Maternal blood samples/Maternal and cord blood samples	[97,98]
FGF21 ^#&^	Maternal	↑	Enhancement carbohydrate and lipid metabolism	Blood samples/Meta-analysis	[101,102]
GHR ^$$^	Placental	↑	Regulation of GH function in growth, metabolism, and other physiological functions	In vitro analysis with *Sprague Dawley* rats embryo samples	[47]
Gluconeogenic AA ^##^ (Glutamine and Glutamate)	Maternal	↑	Aminoacids related to glucose metabolism	Metabolomics (maternal blood and urine samples)	[57,103]
GLUT4 ^##^	Placental	↓	Insulin-regulated glucose uptake transporter	Human placenta samples	[38]
I-CAM-1 ^&&^	Maternal	↑	Cell adhesion molecule related with a proinflammatory state	Prospective case–control study	[58]
IGF-1 ^#$^	Placental/ Maternal	↑/↓	Increased fetal growth	Human placenta samples/Human placenta and maternal blood samples	[38,44,46]
IGFBP-1 ^#$^	Placental	↓	Modulation of biological functions of IGFs	Human placenta	[116]
Insulin ^##^	Maternal	↑	Reduction in blood glucose levels and induction of glucose storage in the liver	In vivo study with rat pancreas	[61]
IR ^##^	Placental	↑	Promotion of cellular metabolism	Human placenta and maternal blood samples	[44]
Irisin ^##^	Maternal	↑	Transformation of white adipose tissue to brown adipose tissue, inducing energy expenditure	Maternal blood, umbilical cord and placenta samples	[127]
IRS-1 (ph) ^##^	Placental	↑	Docking protein for insulin signal transmission	Human placenta samples	[75]
Kisspeptin ^$$^	Maternal	↓	Neuroactive hormone that adapts maternal physiology to pregnancy	Maternal and cord blood and human placenta samples/Review	[107,110]
Leptin ^#&^	Maternal	↑	Suppression of appetite and control of the energy expenditure in white adipose tissue. In pregnancy, maintains increased energy intake for the fetal growth	Maternal blood samples/Human placenta and maternal blood samples	[22,121]
miR-21 ^$$^		↑	micro-RNA in JAK/STAT pathway linked to macrosomic babies.	Human placenta samples/Maternal blood samples/Review	[53,124,125]
miR-675 ^$$^	Maternal		micro-RNA increases placental growth	In vitro analysis (cell line culture) and in vivo analysis (C57/BL6 and H19^Δ3^ transgenic line)/Review	[53,126]
NEFA ^#&^	Maternal	↑	Source of energy for skeletal muscle	Primary cell culture of human placentas	[95]
p-S307 ^##^	Placental	↓	Phosphorylated Ser^307^ of IRS-1, blocks insulin action	Human placenta samples	[38]
p-Y612 ^##^	Placental	↓	Phosphorylated site of IRS-1 intermediates with PI3K activity	Human placenta samples	[38]
p58a ^$$^	Placental	↑	An ER molecular chaperone	Human placenta samples	[75]
pAKT ^##^	Placental	↓/↑	Phosphorylation and regulation of multiple intracellular signaling pathways (metabolism, apoptosis, and proliferation)	Human placenta samples	[38]
pGH ^$$^	Maternal	↑	Activation of maternal and fetal metabolism and growth	Maternal blood samples/Review	[54,115]
PI3K ^##^	Placental	↓	Activation of multiple intracellular signaling pathways (growth, proliferation, migration, secretion, differentiation, transcription, and translation)	Human placenta samples	[38]
Proinflamatory Cytokines ^&&^ (TNF-a, IL-6, IL-1, MCP-1)	Maternal	↑	Activation of multiple cellular-mediated responses	A prospective case–control study	[58]
Resistin ^##^	Placental	↑	Inflammation and impaired glucose tolerance	Human placenta and maternal blood samples	[22]
SNAT1 ^#$^	Placental	↑	Amino acid transporter in the placenta	Human placenta and maternal blood samples	[44]
SOCS2 ^#$&^	Maternal	↓	Inhibition of GH and IGF-1 signaling, thus regulates body growth and development, immune response, lipid and glucose metabolism, and β-cell physiology	Human placenta samples/In vivo evaluation of *Socs2^−/−^* mice	[35,128,129]
Succinic acid ^##^	Maternal	↑	Intermediate product of the Krebs cycle. To produce energy in Krebs cycle	Metabolic plasma analysis (maternal blood samples)	[57,104]
TG ^#&^	Placental/Cord Blood/Maternal	↑/=	Source of energy for the body	Human placenta and maternal blood samples/Maternal and cord blood samples	[85,86,97]
V-CAM-1 ^&&^	Maternal	↑	Cell adhesion molecule related to a proinflammatory state	A prospective case–control study	[58]

Biomarkers associated with gestational diabetes mellitus (GDM) and macrosomia found in the literature described by maternal or placental origin, their up-(↑) or down(↓)regulation or no changes (=)-in blood or tissues, associated function, and study type and evidence. The mechanism of disease that each biomarker is related to is as follows: ^#^ GDM Markers; ^&^ Proinflammatory Markers; ^$^ Macrosomia Markers. Described up-regulated biomarkers in the literature associated with GDM and macrosomia are α-hidroxibutirato (2-Hydroxybutyric), 3-Hydroxybutyric, ANGPL-3-4-8 (angiopoietin-like protein), Rho-guanine nucleotide exchange factor 11 (ARHGEF11), branched-chain amino acids (BCAAs), C-peptide, Estrogen Receptor-α (ER-α), Free Fatty Acids (FFAs) (without clear evidence in the literature), Fibroblast Growth Factor 21 (FGF21), Growth Hormone Receptor (GHR), gluconeogenic amino acids (AAs), Intracellular Adhesion Molecule – 1 (ICAM-1), insulin, insulin receptor (IR), irisin, phosphorylated insulin receptor substrate – 1 (IRS-1 (ph)), leptin, miR-21, miR-675, Non-Esterified Fatty Acids (NEFAs), protein kinase (p58α), Placental Growth Hormone (pGH), proinflammatory cytokines, resistin, Sodium Neutral Amino Acids Transporter (SNAT1), succinic acid, Triglycerides (TGs) (with not clear evidence according to the literature), and Vascular Cell Adhesion Molecule (V-CAM-1). Conversely, down-regulated biomarkers described in the literature as associated with GDM and macrosomia are adiponectin, AMP-activated protein kinase (AMPK), estradiol (E2), fatty acid oxidation (FAO), Glucose Transporter 4 (GLUT4), kisspeptin, phosphorylated Serine 307 (p-S^307^), phosphorylated Tyrosine 612 (p-Y^612^), Fosfatidilinositol-3-Kinasa (PI3K), and suppressor of cytokine signaling 2 (SOCS2). Insulin Growth Factor-1 (IGF-1) and phosphorylated protein kinase B (pAKT) are controversially up- or down-regulated for GDM and macrosomia, depending on the study.

## 7. SOCS2 Mechanism in Gestational Diabetes and Macrosomia

T2D arises when pancreatic β-cells fail to compensate for increased insulin resistance, with cytokines playing a significant role in both insulin and leptin resistance, contributing to β-cell dysfunction [130,131]. The SOCS family, particularly SOCS2, is crucial in these processes. SOCS2 mitigates the adverse effects of proinflammatory cytokines and down-regulates GHR signaling, which can exacerbate insulin resistance by promoting hepatic glucose production [49,132,133,134,135].

SOCS2 is up-regulated in pancreatic islet cells under diabetic-like conditions, resulting in reduced insulin production [133,135]. In experiments with SOCS2-deficient mice, a high-fat diet resulted in impaired glucose tolerance and insulin resistance, while SOCS2 overexpression improved glucose tolerance and insulin sensitivity [7,128,132]. Human genetic studies have linked specific single nucleotide polymorphisms (SNPs) in the SOCS2 gene to T2D prevalence in various populations, suggesting a genetic component to its role in diabetes [6,8] (Figure 5).

The hormonal fluctuations that occur during pregnancy indicate that SOCS2 is an essential regulator of growth, development, and metabolic processes related to gestation (Figure 6). Many actions of these hormones, cytokines, and growth factors are mediated through the JAK/STAT signaling pathway, where SOCS proteins serve as negative regulators. PRLR expression is significantly elevated throughout pregnancy and primarily signals via the JAK/STAT pathway. Specifically, within the mouse ovary during gestation, there is an up-regulation of SOCS2 gene expression triggered by elevated lactogen levels; this acts as a crucial physiological inhibitor for PRLR signaling [16,19,20] and plays a part in pancreatic β-cell proliferation and survival.. Reduced insulin sensitivity during pregnancy is induced by heightened levels of placental and maternal hormones that can interfere with insulin signaling, in which SOCS2 plays a key regulatory role [62,107] (Figure 3).

While embryonic and fetal growth are largely independent of GH activity, SOCS2 regulates GH signaling during postnatal growth [55]. Elevated SOCS2 levels in placentas of fetuses with IUGR indicate its role in controlling insulin sensitivity and inhibiting GH activity [35]. The deletion of SOCS2 in mice leads to gigantism, correlating with increased IGF-1 levels and fetal overgrowth, suggesting that SOCS2 acts as a negative regulator of the growth axis [7,15,128,129] (Figure 3).

Sex steroid E2 induces SOCS2, down-regulating GHR-mediated signal transduction [113,136]. Such interactions may be associated with hepatic metabolic changes that promote energy homeostasis, enhance insulin sensitivity, improve β-cell function, reduce inflammation, and optimize body fat mass distribution [137,138]. SOCS2 is believed to down-regulate pancreatic β-cell proliferation since SOCS2 gene expression increases during pregnancy [16,139]. Additionally, gonadotropins and E2 regulate activity mediated by IGF-1, which acts as a downstream mediator for GH [47].

SOCS2 is involved in GDM, as SOCS2-knockout mice exhibit hyperglycemia during pregnancy [129,140] (Figure 3). Despite its regulatory role in the IGF-1 axis and hepatic metabolism, the precise mechanisms by which SOCS2 influences GDM and fetal growth require further investigation. Overall, SOCS2 interacts with metabolic organs and tissues, potentially affecting signaling pathways related to both GDM and fetal overgrowth, while also playing a critical role in insulin sensitivity and β-cell function in T2D.

## 8. Limitations, Directions for Future Research, and General Conclusions

Despite advances in understanding GDM and its complications, gaps remain in our knowledge of specific biomarkers. SOCS2, critical for cytokine signaling and metabolic regulation, presents challenges in interpreting study results.

Firstly, differentiating between markers specific to GDM and those associated with macrosomia is difficult. Macrosomia can result from GDM or other factors, making it unclear whether certain biomarkers indicate maternal hyperglycemia or disruptions in growth signaling pathways.

Secondly, comparing rodent models to human studies introduces limitations due to species-specific differences. Research findings vary widely, contributing to inconsistencies.

Future clinical studies should measure and compare biomarker levels in GDM and non-GDM pregnancies and contrast findings with macrosomia. This would elucidate molecular mechanisms linking SOCS2 with glucose metabolism and fetal growth patterns, enhancing our understanding of normal pregnancy mechanisms and implications for GDM pathophysiology and macrosomia development. Furthermore, the pharmacological modulation of SOCS2, currently achievable through commercial drugs, presents a promising avenue for the future targeted therapies. Prospective cohort studies to evaluate its predictive value for GDM and macrosomia, as well as interventional studies to assess the efficacy of SOCS2-modulating drugs in preventing or managing these conditions, would be valuable for further investigation.

Identifying reliable biomarkers for GDM and macrosomia is crucial for early diagnosis, effective management, and improved outcomes. This review explores potential biomarkers associated with inflammation, growth, and metabolism, providing insights into the pathophysiology of GDM and macrosomia. Understanding these biomarkers can enhance predictive capabilities and develop targeted interventions to mitigate adverse effects.

SOCS2 appears to have an indirect effect during pregnancy that might be disrupted in GDM. Data indicate crosstalk between pathogenic pathways known for GDM and macrosomia and signaling pathways involving SOCS2. Alterations in serum and placental biomarkers related to GDM and macrosomia may be associated with SOCS2 dysfunction. While few studies directly link SOCS2 to pregnancy, glucose, lipid metabolism, and fetal overgrowth, the existing evidence suggests a significant role for SOCS2 in these processes. Further investigation into maternal-fetal interaction mechanisms through the placenta is warranted to confirm the serum and placental markers associated with protection against GDM and macrosomia.

## Figures and Tables

**Figure 1 nutrients-17-01519-f001:**
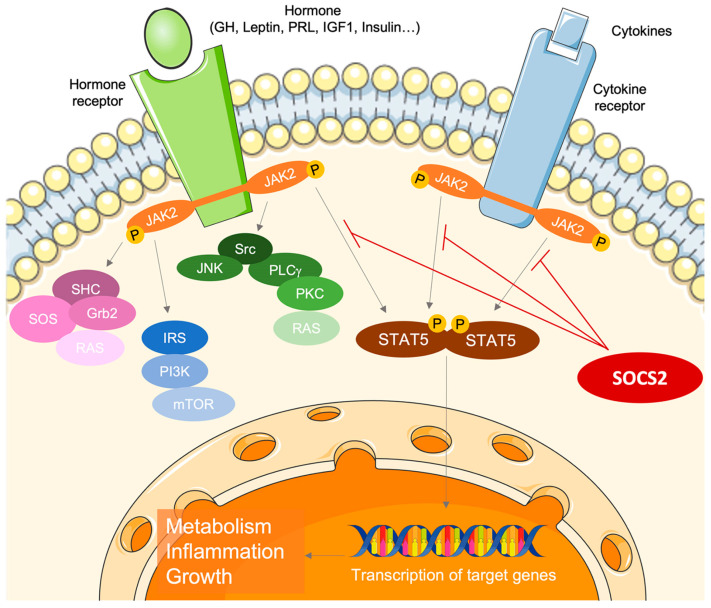
The role of suppressor of cytokine signaling 2 (SOCS2) as a key regulator of cytokine and growth factor signaling through the Janus kinase (JAK)/signal transducers and activators of transcription (STAT) pathway.

**Figure 2 nutrients-17-01519-f002:**
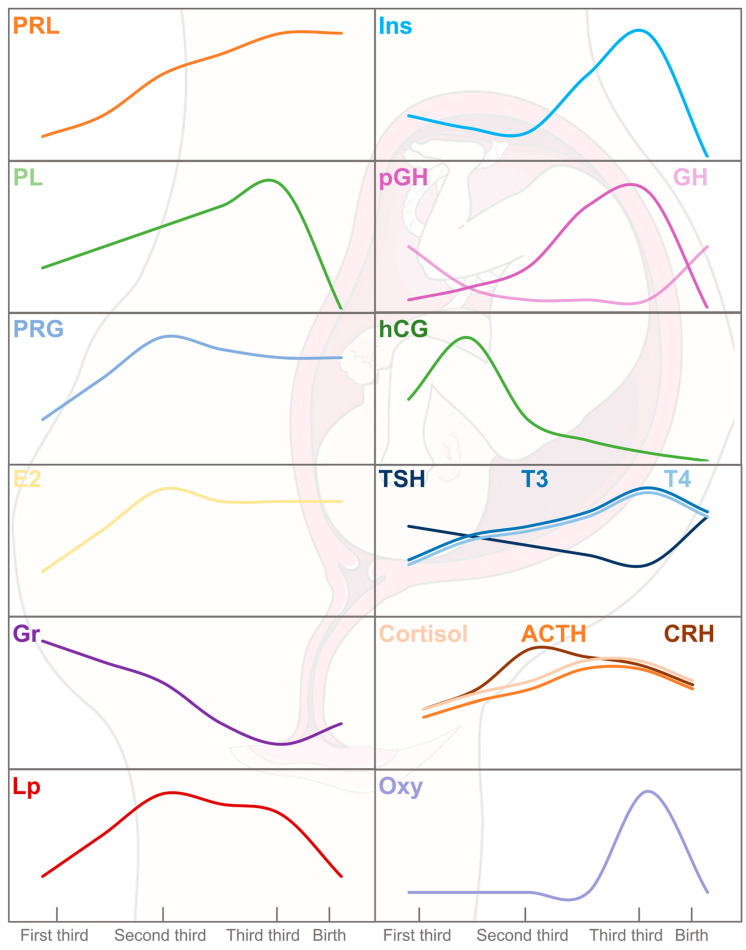
Hormone physiology during gestation period and labor. Insulin (Ins), prolactin (PRL), progesterone (PRG), estradiol (E2), growth hormone (GH), thyroid-stimulating hormone (TSH), triiodothyronine (T3), thyrodine (T4), corticotropin (CRH), adrenocorticotropic hormone (ACTH), cortisol (Cor), and oxytocin (Oxy) are of maternal origin. In addition, placental lactogen (PL), leptin (Lp), ghrelin (Gr), gonadotropin (hCG), and corticotropin (CRH) are produced by the placenta. The serum levels of PRL, E2, Ins, PL, Lp, CRH, ACTH, and Cor are increased from the first to the third trimester of pregnancy followed by a decrease or stabilization during labor. However, an opposite kinetic can be observed for GH and hCG. In the fetus, pituitary GH is replaced by placental GH (pGH). After birth, pituitary GH secretion increases until it reaches maximal serum levels at puberty.

**Figure 3 nutrients-17-01519-f003:**
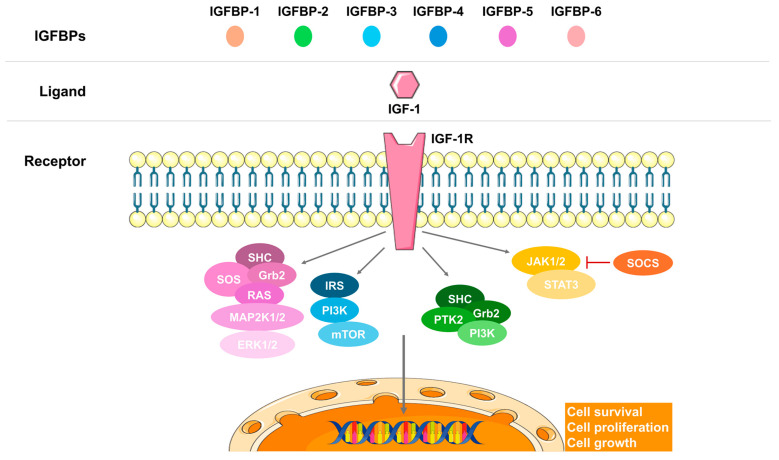
The insulin growth factor 1 (IGF-1) signaling axis. The IGF-binding proteins (IGFBPs) facilitate the binding of the ligand IGF-1 to its receptor (IGF-1R), transducing the signal through various pathways: the mitogen-activated protein kinase (MAPK), the mammalian target of rapamycin (mTOR), phosphoinositide 3-kinase (PI3K), or the Janus kinase (JAK)/signal transducer and activator of transcription (STAT). These pathways lead to the expression of target genes involved in cell survival, proliferation, and growth.

**Figure 4 nutrients-17-01519-f004:**
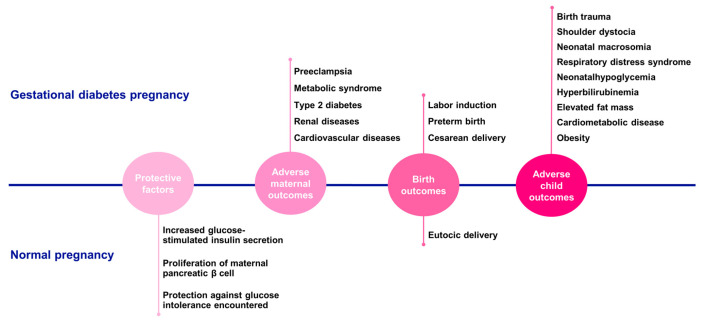
Gestational diabetes pregnancies vs. normal pregnancies. Comparison of the main findings regarding protective factors, adverse maternal outcomes, birth outcomes, and adverse child outcomes between gestational diabetes and normal pregnancies.

**Figure 5 nutrients-17-01519-f005:**
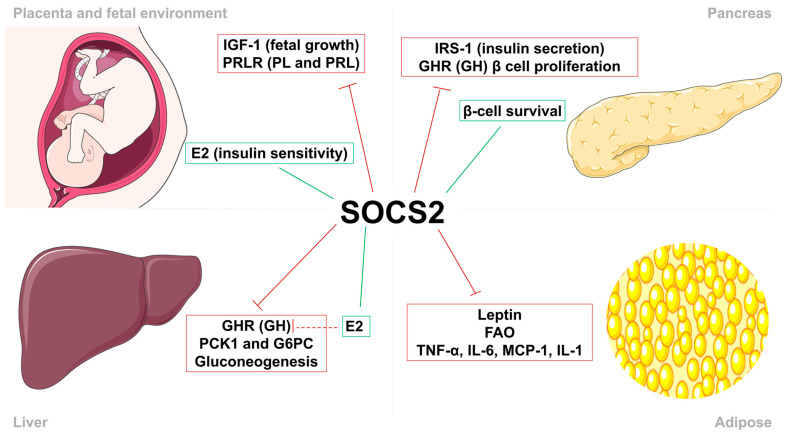
Suppressor of cytokine signaling 2 (SOCS2) regulates insulin sensitivity and body growth. SOCS2 stimulation (green) and inhibition (red) of metabolic processes, growth factors, cytokines and hormones in the feto-placental unit (FPU), liver, pancreas, and adipose tissue. In the FPU, SOCS2 regulates fetal growth by inhibiting insulin growth factor–1 (IGF-1), placental lactogen (PL), and prolactin (PRL), and increases insulin sensitivity by inducing estradiol (E2) synthesis. Metabolic actions in the pancreas consist of the reduction in insulin secretion by inhibiting insulin receptor substrate–1 (IRS-1) activity and β-cell proliferation through growth hormone (GH). Moreover, SOCS2 directly promotes the survival of these pancreatic β-cells. In the liver, SOCS2 reduces gluconeogenesis through direct and indirect mechanisms. Finally, SOCS2 regulates proinflammatory processes by inhibiting the action of leptin, fatty acid oxidation (FAO), and proinflammatory cytokines in adipose tissue.

**Figure 6 nutrients-17-01519-f006:**
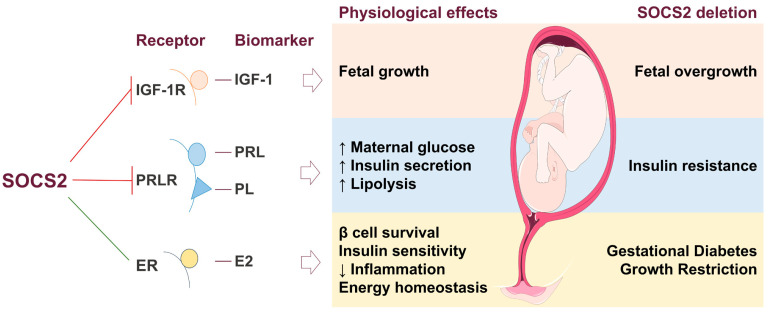
Suppressor of cytokine signaling 2 (SOCS2) in the feto-placental unit (FPU) and effects on fetal metabolism. The inhibition (red line) and stimulation (green line) of biomarkers and their related metabolic processes in the FPU by SOCS2. SOCS2 regulates fetal growth by inhibiting insulin-like growth factor 1 (IGF-1) through its receptor (IGF-1R). The prolactin receptor (PRLR) is negatively regulated by SOCS2, which increases maternal glucose, insulin secretion, and lipolysis induced by prolactin (PRL) and placental lactogen (PL) hormones. Estradiol (E2) is stimulated by SOCS2, enhancing energy homeostasis by promoting β-cell survival, improving insulin sensitivity, and reducing inflammation. In the absence of SOCS2, IGF-1, PRL, and PL are overexpressed, leading to fetal overgrowth and insulin resistance. Conversely, reduced levels of E2 result in GDM and growth restriction.

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
