# Peer review of "Mechanisms of Fetal Overgrowth in Gestational Diabetes: The Potential Role of SOCS2"

_nutrients, 2025, doi:10.3390/nu17091519_

Round 1

Reviewer 1 Report

Comments and Suggestions for Authors

This manuscript is well-designed, appropriately prepared and has an actual scientific role. The authors also did a good bibliographic search. Considering the increasing evidence of the relationship between gestational diabetes and fetal overgrowth, this review manuscript aims to provide a comprehensive analysis of the mechanisms involved in this process, with a final focus on the potential role of SOCS2.  

''GDM is a common pregnancy complication associated with an increased risk of adverse pregnancy outcomes such as preeclampsia, macrosomia, or neonatal hypoglycemia. The manuscript highlights the underlying mechanisms regarding GDM and macrosomia, starting from a comprehensive and conclusive subchapter in which the authors describe the endocrine and metabolic changes that occur during pregnancy. Additionally, biomarkers may offer a new avenue for improved prediction of fetal macrosomia, although current evidence in this emerging field is still partially deciphered. In this regard, the authors provide an up-to-date review (in subchapter 6) of the placental and maternal blood biomarkers identified in the literature related to GDM and macrosomia.  

Although SOCS2 regulates hepatic gluconeogenesis via the JAK2-STAT5 pathway and is considered a potential therapeutic target for diabetes treatment, few studies have explored the underlying mechanisms linking SOCS2 to GDM and fetal macrosomia. In conclusion, this manuscript summarises the complex interactions between GDM and macrosomia, highlighting the interplay between the pathogenic pathways involved in both conditions and the signalling axis mediated by SOCS2. The authors cite a wide range of studies to support their claims, making the review scientifically robust. ''  

To increase the quality of the manuscript, I have only a few suggestions for the authors: 

Kindly increase the typographical quality of the Figures (all Figures)  

Please revise the references list according to the Nutrients journal recommendations (please see references 97, 113, and 12).

Author Response

Comment 1: Kindly increase the typographical quality of the Figures (all Figures)  

Response: Thank you for highlighting this issue. In accordance with the reviewer's comments, we have enhanced the typographical quality of all figures, including those originally submitted and the new ones prepared in response to the reviewers' feedback.

Comment 2: Please revise the references list according to the Nutrients journal recommendations (please see references 97, 113, and 12).

Response: We agree with the reviewer's suggestion. All references have been updated to conform to the 'Nutrients' style using Mendeley citation settings

Reviewer 2 Report

Comments and Suggestions for Authors

Manuscript Title

Mechanisms of fetal overgrowth in gestational diabetes: the potential role of SOCS2

General Assessment:

This is a comprehensive and well-structured review that delves into the role of SOCS2 in gestational diabetes mellitus (GDM) and fetal macrosomia. The authors provide an extensive overview of metabolic and hormonal changes during pregnancy, associated biomarkers, and the mechanistic role of SOCS2. The manuscript is timely and potentially impactful, given the increasing prevalence of GDM and associated perinatal complications.

  1. Abstract

Evaluation:
The abstract provides a clear and concise overview of the manuscript, outlining the key issues and the proposed role of SOCS2 in the pathophysiology of GDM and macrosomia.

Suggestions:

  • Include specific examples of key cytokines or hormones to strengthen clarity.
  • Explicitly state the novelty of the hypothesis regarding SOCS2.
  1. Introduction

Evaluation:
The introduction offers a thorough background on SOCS2 and its role in cytokine signaling, particularly within the JAK/STAT pathway. The rationale for focusing on SOCS2 in the context of GDM and fetal overgrowth is logically developed.

Strengths:

  • Justifies the need for the review.
  • Highlights gaps in current knowledge.

Suggestions:

  • Clarify the hypothesis by succinctly linking SOCS2 dysfunction to clinical consequences (e.g., insulin resistance, fetal macrosomia).
  • Consider summarizing complex background details into a conceptual diagram or pathway overview.
  1. Endocrine-Metabolic Sketch of Pregnancy

Evaluation:
This section is comprehensive, well-cited, and logically organized. It effectively outlines the dynamic hormonal adaptations during pregnancy.

Strengths:

  • Strong integration of endocrine and metabolic pathways.
  • Clear explanation of maternal-fetal nutrient partitioning.

Suggestions:

  • Condense some physiological descriptions to reduce redundancy and maintain focus on relevance to GDM and SOCS2.
  1. Embryonic and Fetal Growth

Evaluation:
This section links endocrine regulation to sexual dimorphism in fetal growth and introduces IGF-1 and mTOR pathways.

Strengths:

  • Highlights the complexity of growth regulation.
  • Addresses sex-specific differences in fetal responses.

Suggestions:

  • Clarify the interplay between fetal and maternal IGF-1 and its limitations due to the placental barrier.
  • Include a graphical summary of signaling cascades.
  1. Gestational Diabetes

Evaluation:
The manuscript provides an excellent overview of GDM pathophysiology and its impact on both mother and fetus.

Strengths:

  • Incorporates inflammatory, metabolic, and hormonal components.
  • Cites animal models and human data appropriately.

Suggestions:

  • Discuss the limitations of rodent models more explicitly.
  • Consider integrating a timeline or schematic comparing normal vs. GDM pregnancies.
  1. Macrosomia

Evaluation:
The section successfully connects maternal GDM to fetal overgrowth mechanisms.

Strengths:

  • Rich discussion of nutrient and hormonal mediators.
  • Balanced coverage of short- and long-term complications.

Suggestions:

  • Some repetition from earlier sections could be trimmed.
  • Include more discussion on fetal programming and long-term epigenetic changes.
  1. Biomarkers of GDM and Macrosomia

Evaluation:
This section is a strong contribution, presenting an extensive list of maternal and placental biomarkers with clear evidence mapping.

Strengths:

  • Detailed and up-to-date review of literature.
  • Table 1 is a valuable resource for readers.

Suggestions:

  • Consider summarizing key biomarkers by function (inflammatory, metabolic, growth) to enhance clarity.
  • Indicate clinical applicability or diagnostic potential of top candidates.
  1. SOCS2 Mechanism in GDM and Macrosomia

Evaluation:
This is the core of the manuscript and effectively elaborates on the SOCS2 hypothesis.

Strengths:

  • Integrates data from animal models, human studies, and molecular mechanisms.
  • Provides novel insights.

Suggestions:

  • Some mechanistic descriptions could benefit from simplification for accessibility.
  • The figures (Fig. 2 & 3) are excellent but could include more quantitative markers (e.g., expression levels, changes in SOCS2).
  1. Limitations, Future Directions, and Conclusions

Evaluation:
This section appropriately addresses existing research gaps and suggests future directions.

Strengths:

  • Acknowledges translational challenges.
  • Proposes specific next steps.

Suggestions:

  • Emphasize the clinical potential of SOCS2-based diagnostics or therapies.
  • Consider proposing prospective cohort or interventional study designs.

Figures and Tables

Evaluation:
High quality and well-integrated into the text. Table 1 is particularly valuable.

Suggestions:

  • Ensure all acronyms are defined in legends.
  • Enhance contrast or clarity in schematic diagrams for better visual impact.

References

Evaluation:
The reference list is current and relevant.

Suggestions:

  • Ensure consistency in formatting.
  • Double-check for duplication or outdated citations.

Author Response

Comment 1: Include specific examples of key cytokines or hormones to strengthen clarity.

Response: We have accordingly included relevant examples of cytokines and hormones in the abstract, which can be found in lines 18 and 19 of this section (page 1): “Several cytokines and hormones, such as interleukin 6 (IL-6), insulin growth factor 1 (IGF-1), prolactin (PRL), or progesterone, are essential for fetal growth, control of in-flammatory response, and regulation of lipid and carbohydrate metabolism to meet energy demands during pregnancy.”

Comment 2: Explicitly state the novelty of the hypothesis regarding SOCS2.

Response: We appreciate your suggestion to highlight the novelty of our hypothesis. We have updated the end of the abstract to clearly state our hypothesis regarding SOCS2 mechanisms involved in GDM and macrosomia. You will find these changes in lines 25-27 (page 1): “Here we review this pathophysiology and pose the hypothesis that an aberrant response to cytokine receptor activation, particularly involving suppressor of cytokine signaling 2 (SOCS2), contributes to GDM and fetal macrosomia. This novel perspective suggests an unexplored mechanism by which SOCS2 dysregulation could impact pregnancy outcomes.

Comment 3: Clarify the hypothesis by succinctly linking SOCS2 dysfunction to clinical consequences (e.g., insulin resistance, fetal macrosomia).

Response: Thank you for your suggestion. We have added a sentence at the end of the introduction (lines 60-63 on page 2) to link SOCS2 dysfunction with clinical consequences: “We also propose the SOCS2 protein as a focus for research into the physiology and physiopathology of pregnancy, as dysregulation in its signaling pathway may lead to clinical consequences, such as insulin resistance and fetal overgrowth.” 

Comment 4: Consider summarizing complex background details into a conceptual diagram or pathway overview.

Response: According to your suggestion, we have created a schematic summary of the JAK/STAT pathway, including the role of SOCS2. This is now Figure 1, mentioned in line 43 (page 2), and appears at the end of the section (lines 64-65) on page 2. Caption to Figure 1 (lines 66-68, page 2): “Figure 1. The role of Suppressor of Cytokine Signaling 2 (SOCS2) as a key regulator of cytokine and growth factor signaling through the Janus Kinase (JAK)/ Signal transducers and activators of transcription (STAT) pathway.”.

Comment 5: Condense some physiological descriptions to reduce redundancy and maintain focus on relevance to GDM and SOCS2.

Response: In agreement with the reviewer comment, we have removed some physiological descriptions that were not represented in the original Figure 1 (now Figure 2). Strikethrough text: lines 110-113, page 3: Progesterone and estrogens regulate food intake in murine models during pregnancy. Progesterone increases food consumption and weight gain, while estrogens interact with hypothalamic neurons via estrogen receptors (ER) to prevent fat accumulation, reduce food intake, and enhance energy expenditure[17]; Lines 127-128, page 4: When maternal glucose levels are low, pGH levels tend to rise, and when glucose levels are high, pGH levels fall., lines 146-147, page 4: Other metabolites like deoxycorticosterone (DOC) and corticosteroid-binding globulin (CBG) are elevated during pregnancy[31]. Additionally, we have included the disrupted adaptations for GDM in lines 94-95 (page 3): “However, in GDM pregnancies, these adaptations may be disrupted, leading to adverse pregnancy outcomes.”. Moreover, at the end of the section (lines 153-156 on page 4), we elucidate the link between hormone receptors, their signaling pathways, and the role of SOCS2 in normal pregnancy physiology: “Several of the main hormones involved in pregnancy, such as PRL, PL, progesterone, estrogens, GH, pGH, IGF-1, insulin, and leptin, are transduced by the JAK/STAT pathway. Therefore, SOCS2, as key regulator of this pathway, may be involved in the endocrine and metabolic adaptations during pregnancy [1,3,33]”.

Comment 6: Clarify the interplay between fetal and maternal IGF-1 and its limitations due to the placental barrier.

Response: Thank you for pointing this out. In the “Embryonic and fetal growth” section, we have included a paragraph on lines 203-210, page 6, trying to clarify the interplay between maternal and fetal IGF-1 and its limitations due to the placental barrier. We have added relevant reference to write this paragraph (50, 51 and 52):The interplay between fetal and maternal IGF-1 is indeed crucial for fetal development. Maternal IGF-1 facilitates nutrient transfer to the fetus, supporting its growth and development. Conversely, fetal IGF-1 drives growth by promoting anabolic processes and DNA synthesis[50,51]. The placental barrier plays a significant role in regulating this process. By expressing IGF-1 receptors and IGF binding proteins (IGFBPs), the placenta modulates the availability and function of IGF-1 between the mother and fetus[50,52]. This regulation is essential to ensure proper nutrient delivery and growth signals, although it can also pose limitations when placental function is compromised.”

Comment 7: Include a graphical summary of signaling cascades.

Response: following the reviewer’s suggestion, we have included a new graphical summary of the IGF-1 signaling in the embryonic and fetal growth section. New Figure 3 that we mention on line 187, page 6, and place on page 6, line 190. Caption to Figure 3 (lines 191-196, page 6): “Figure 3. The insulin growth factor 1 (IGF-1) signaling axis. The IGF binding proteins (IGFBPs) facilitate the binding of the ligand IGF-1 to its receptor (IGF-1R), transducing the signal through various pathways: the mitogen-activated protein kinase (MAPK), the mammalian target of rapamycin (mTOR), phosphoinositide 3-kinase (PI3K), or the Janus kinase (JAK)/signal transducer and activator of transcription (STAT). These pathways lead to the expression of target genes involved in cell survival, proliferation, and growth.”.

Comment 8: Discuss the limitations of rodent models more explicitly.

Response: Thank you for your suggestion. In agreement with the reviewer’s comment, we have included a paragraph discussing the limitations of using animal models in fetal development and gestation. We have also included a new reference, number 63. This new paragraph is on page 8 lines 250 to 257. “Animal model studies are essential for understanding the mechanisms of fetal development and gestation, but they have significant limitations. The placental interface differs between species; in humans, maternal blood directly contacts a single layer of trophoblasts in the chorion plate, whereas in rodents, the placental barrier is hemotrichorial, consisting of three layers of trophoblasts. Additionally, rodent pregnancies are typically multiparous, and their gestation period is much shorter than that of humans. Therefore, rodent models are valuable for studying basic biological processes, but their translatability to human pregnancies should be carefully considered[63].

Comment 9: Consider integrating a timeline or schematic comparing normal vs. GDM pregnancies.

Response: Thank you for your recommendation. In agreement, we have included a timeline comparing gestational diabetes pregnancies and normal pregnancies (mentioned on line 230, page 7). This is the new Figure 4 that is now mentioned on line 230, page 7 and placed on page 7 line 232. Caption to Figure 4 (lines 233-235, page 7): “Figure 4. Gestational diabetes pregnancies vs normal pregnancies. Comparison of the main findings regarding protective factors, adverse maternal outcomes, birth outcomes, and adverse child outcomes between gestational diabetes and normal pregnancies.

Comment 10: Some repetition from earlier sections could be trimmed.

Response: Thank you for your valuable feedback. We understand your concern regarding the repetition in the discussion of maternal and fetal complications. The similarities in these sections arise because gestational diabetes and macrosomia are closely related but distinct pathologies. Gestational diabetes often leads to macrosomia, resulting in overlapping maternal and fetal complications. However, each condition has unique aspects that warrant separate discussion to fully address their implications. However, we have trimmed the following text (lines 301-302, page 9): While fetal macrosomia is commonly associated with GDM, it can also be linked to advanced maternal age, excessive weight gain during pregnancy, or pre-existing obesity.

Comment 11: Include more discussion on fetal programming and long-term epigenetic changes.

Response: in according to your valuable point, we have included a new paragraph in the macrosomia section discussing about epigenetics changes during fetal programing affecting adult offspring. New references, 92 and 93, have been used and added to write this new paragraph (lines 370-382, page 10: “Fetal metabolic programming caused by an adverse intrauterine environment can lead to metabolic syndrome in adult offspring. Although the precise mechanisms remain incompletely understood, epigenetic programming is proposed as a critical underlying factor. This process influences gene expression and cellular function through stable epigenetic modifications that do not alter the DNA sequence. These modifications can be maintained through mitosis and transmitted across generations, thereby contributing to metabolic syndrome in adults[92]. Genes associated with adipose tissue development, hyperphagia, energy balance regulation, lipid metabolism, insulin signaling, adipocytokines, proinflammatory factors, pancreatic islet beta cell development, and DNA or histone-modifying enzymes can undergo changes due to fetal epigenetic modifications. This contributes to metabolic syndrome in adults. Notable genes affected include IGF1R, IGF2, FGF21, ER, Glut4, IGFBP1, leptin, and adiponectin[51,92,93].”.

Comment 12: Consider summarizing key biomarkers by function (inflammatory, metabolic, growth) to enhance clarity.

Response: Thank you for your comment. As it was also suggested by reviewer 3, we have divided section 6 “Biomarkers of gestational diabetes and macrosomia” in two new subsections, titled as follows: “6.1. Gestational diabetes biomarkers” (pages 11-12) and “6.2. Macrosomia biomarkers” (pages 13-14). The classification by inflammation, metabolism or growth factors is already included in Table 1. You can find this new section on pages 10 to 14.

Comment 13: Indicate clinical applicability or diagnostic potential of top candidates.

Response: According to your suggestion, we clarify, at the end of the “Biomarkers of gestational diabetes and macrosomia biomarkers” section, the potential use of the top biomarkers to diagnose GDM and macrosomia. The new paragraph is on page 14 lines 548-554. The clinical applicability of biomarkers for gestational diabetes mellitus (GDM) and macrosomia holds promise for enhancing prenatal care and improving pregnancy outcomes. Given the challenges in accessing the placenta for diagnosing GDM or macrosomia during pregnancy, maternal serum biomarkers emerge as potential diagnostic candidates for these conditions. Biomarkers such as C-peptide, adiponectin, and insulin-like growth factor 1 (IGF-1) provide valuable insights into fetal growth patterns associated with GDM.”

Comment 14: Some mechanistic descriptions could benefit from simplification for accessibility.

Response: In agreement with your suggestion, we have simplified some of the mechanistic descriptions to enhance accessibility. We have trimmed complex molecular details and replaced them with more accessible terminology. You can find these changes along section 7 “SOCS2 mechanism in gestational diabetes and macrosomia”.

Comment 15: The figures (Fig. 2 & 3) are excellent but could include more quantitative markers (e.g., expression levels, changes in SOCS2).

Response: Thank you for your positive feedback on Figures 2 and 3. We appreciate your suggestion to include more quantitative markers. However, we would like to note that Table 1 already addresses changes in expression levels, and the figures illustrate the effects of the presence or absence of SOCS2.

Comment 16 and 17: Emphasize the clinical potential of SOCS2-based diagnostics or therapies. Consider proposing prospective cohort or interventional study designs.

Response: We agree with the reviewer’s comments and have included a paragraph highlighting the clinical potential of SOCS2. Additionally, we propose prospective cohort and interventional studies to further emphasize its significance. This new paragraph is located on page 22 from lines 666 to 671 in “Limitations, directions for future research and general conclusions”: Furthermore, the pharmacological modulation of SOCS2, currently achievable through commercial drugs, presents a promising avenue for future targeted therapies. Prospective cohort studies to evaluate its predictive value for GDM and macrosomia, as well as interventional studies to assess the efficacy of SOCS2-modulating drugs in preventing or managing these conditions, would be valuable for further investigation.”

Comment 18: Ensure all acronyms are defined in legends.

Response: All acronyms have been reviewed in their legends.

Comment 19: Enhance contrast or clarity in schematic diagrams for better visual impact.

Response: All figures, included the originally submitted and the new ones that we have prepared for this revised version of our manuscript, have been corrected to get more quality.

Comment 20 and 21: Ensure consistency in formatting. Double-check for duplication or outdated citations.

Response: All references have been updated to the "Nutrients" style using Mendeley citation settings and reviewed for duplications and completeness.

Reviewer 3 Report

Comments and Suggestions for Authors

A principal limitation of the present review lies in the difficulty of clearly distinguishing between biomarkers specifically associated with gestational diabetes mellitus (GDM) and those implicated in fetal macrosomia. Given that macrosomia may arise independently of maternal hyperglycemia, the attribution of certain biomolecular signals to either condition remains challenging and may confound interpretation of underlying mechanisms.

Furthermore, translational limitations exist due to species-specific differences between rodent models and human physiology. While animal studies offer mechanistic insights, their applicability to human gestation must be interpreted with caution. The variability in findings across studies also underscores the need for standardized methodologies and larger, well-controlled human cohorts to validate the role of SOCS2 in the pathophysiology of GDM and macrosomia.

Recommendation

The manuscript addresses a highly relevant and timely topic in the field of maternal-fetal medicine, providing a comprehensive and well-structured review of the potential role of SOCS2 in the pathophysiology of gestational diabetes mellitus (GDM) and fetal macrosomia. The authors successfully integrate current knowledge on metabolic, endocrine, and inflammatory mechanisms, and propose a novel perspective on SOCS2 as a regulatory node in these processes.

Despite some limitations, particularly regarding the differentiation of GDM-specific versus macrosomia-specific biomarkers and translational constraints from animal models, the review is thorough, well-referenced, and offers valuable directions for future research.

I recommend minor revisions, particularly to further clarify the distinction between GDM and macrosomia-related biomarkers and to more explicitly highlight the translational relevance of the findings to clinical practice.

Author Response

Comment 1: clarify the distinction between GDM and macrosomia-related biomarkers

Response: Thank you for your valuable suggestion. We have divided section 6 “Biomarkers of gestational diabetes and macrosomia” in two new subsections, titled as follows: “6.1. Gestational diabetes biomarkers” (pages 11-12) and “6.2. Macrosomia biomarkers” (pages 13-14). These subsections distinguish between maternal and placental origins for both conditions. We hope these changes make the text clearer. You can find this new section on pages 10 to 14.

Comment 2: Explicitly highlight the translational relevance of the findings to clinical practice.

Response: As it was also highlighted by Reviewer 2, we have included a paragraph in section 8, "Limitations, Directions for Future Research, and General Conclusions," discussing the clinical potential of SOCS2. We propose prospective cohort and interventional studies to support its translational relevance to clinical practice. This new paragraph is on page 22, lines 666-671: Furthermore, the pharmacological modulation of SOCS2, currently achievable through commercial drugs, presents a promising avenue for future targeted therapies. Prospective cohort studies to evaluate its predictive value for GDM and macrosomia, as well as interventional studies to assess the efficacy of SOCS2-modulating drugs in preventing or managing these conditions, would be valuable for further investigation.”